# Archaeometric Characterization of Wall Paintings from Isera and Ventotene Roman Villas

**Paolo Ferretti [1,*], Michela Canali [2] and Barbara Maurina [2,*]**

1    MUSE—Museo delle Scienze di Trento, 38122 Trento, Italy
2    Fondazione Museo Civico di Rovereto, 38068 Rovereto, Italy
*    Correspondence: paolo.ferretti@muse.it (P.F.); maurinabarbara@fondazionemcr.it (B.M.)

**Abstract:** The authors present the first results of an archaeometric research project set up by the Rovereto Civic Museum Foundation in collaboration with MUSE–Trento Science Museum, aiming at analysing and comparing Roman plasters from different sites in order to highlight similarities and differences related to the preparation and realization of Roman wall renderings. The data concern the characterization of plaster samples from the Roman Villa of Isera and Ventotene (northern and central Italy) by means of a thin-section mineropetrographic examination under an optical microscope and a scanning electron microscope (SEM).

**Keywords:** Isera; Ventotene; roman villas; wall plaster; pigments; archaeometry; thin sections

## 1. Introduction

In the Roman period, the use of wall renderings for protective and decorative purposes was widespread. Most of the walls of the houses had painted decoration, usually made with the "fresco" technique, which consists of applying pigments on the lime mortar before it sets. The renderings were carefully prepared by overlapping several layers of mortar. In the treatise *De Architectura* (VII, 3), Vitruvius recommends seven successive layers of three different qualities: a rough layer, three layers of mortar made with sand, and three layers composed of lime and powdered marble. Pliny the Elder in his *Naturalis Historia* (XXXVI, 176) recommends only five layers: three made with sand and two with powdered marble. The archaeological evidence shows that such good renderings are rarely documented in the Roman buildings, and also, other types of aggregate were used.

For a long period in the 19th century, Roman renderings were studied above all, focusing on painted decorations and on stylistic issues, whereas less attention was given to the technique of preparation of mortars and to the characterisation of materials used; however, there were significant exceptions, such as, for instance, the study of Alix Barbet and Claudine Allag, still very relevant today [1], or that of Michel Frizot, a pioneer of archaeometric research on Roman plasters [2].

In recent decades, the archaeometric analyses have acquired an increasing importance in archaeological studies and have become more and more varied and precise, providing a fundamental contribution to the dating and characterization of all types of archaeological finds, including wall plaster. As a matter of fact, the new techniques and technologies allow us to identify the composition of the mortars and the nature of the pigments. This offers a considerable amount of information about the materials used by the ancient painters and the processes implemented in the execution of the renderings and painted decorations, as the recent studies relating to northern and central Italy sites reveal [3–7].

In 2021, the Civic Museum of the Rovereto Foundation, in collaboration with the MUSE–Science Museum of Trento, set up an archaeometric research project in order to carry out a comparative study of samples of wall plaster from different parts of Roman Italy using a technical approach. The purpose of the study is to compare plasters of different origins and chronology in order to understand the evolution of the technique of wall

plasters in the Roman age; the work is in progress and the collection of samples is still ongoing. At the moment, fragments from two Roman villas located in central and northern Italy, at Isera (TN) [8] and at Ventotene (LT) [9] (Figure 1), are under analysis. These villas have been selected because they are both built in the Augustan age and present significant evidence of fragmentary mural paintings dating back to the early imperial age and, in particular, to the third Pompeian Style. The comparison between finds coming from the two sites is interesting in that it permits the verification of technical features and materials used in distant places of Italy in the same period; the results of the research, when compared with data relating to other archaeological sites, may contribute to the knowledge on the methods of realisation of wall paintings in the Roman period.

B.M.

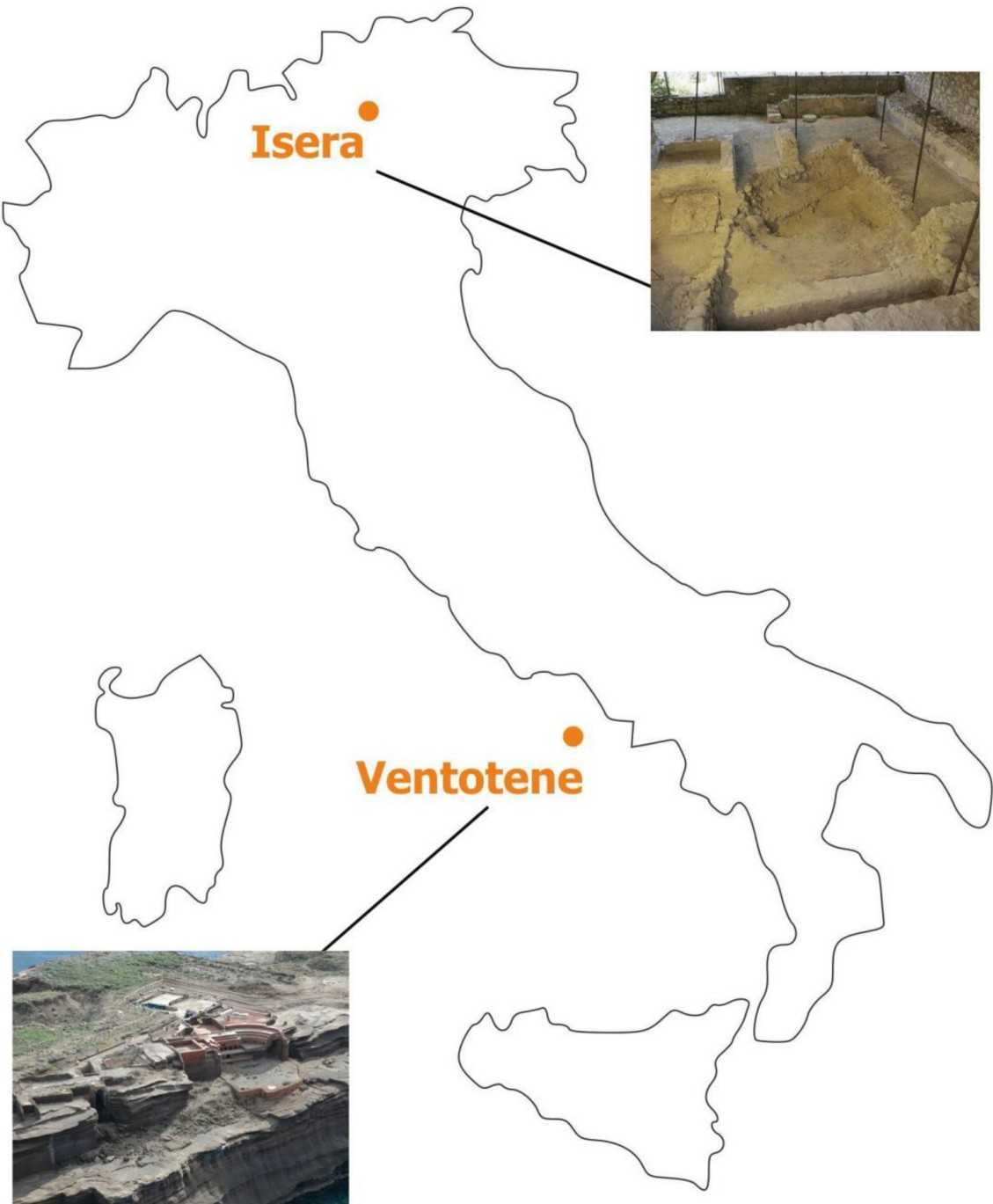

**Figure 1.** Localization of the archaeological sites.

## 2. Materials and Methods

Here, the characterisation of six samples of painted mortar is presented; three of them are from the Isera Roman Villa (IS 6934, 8339 and 8393) and are ascribable to two walls and a ceiling; the remaining three are from Ventotene (VNT 39, 19 and D) and are samples of painted wall plaster, one of stucco decoration and one of ceiling covering. All of them come from the *pars urbana* of the villas.

Six petrographic thin sections were obtained from the plaster samples (Figure 2). The preparation included a vacuum impregnation with fluid epoxy, the cut of the samples, a second impregnation to stabilise the surface, grounding on the lapping machine using 800-grit silicon carbide and a washing with ultrasound to eliminate traces of abrasive from the surface. The samples were finally placed on top of glass slides, cut with a 1 mm diamond saw and ground to 30 µm on a lapping machine with 800-grit silicon carbide.

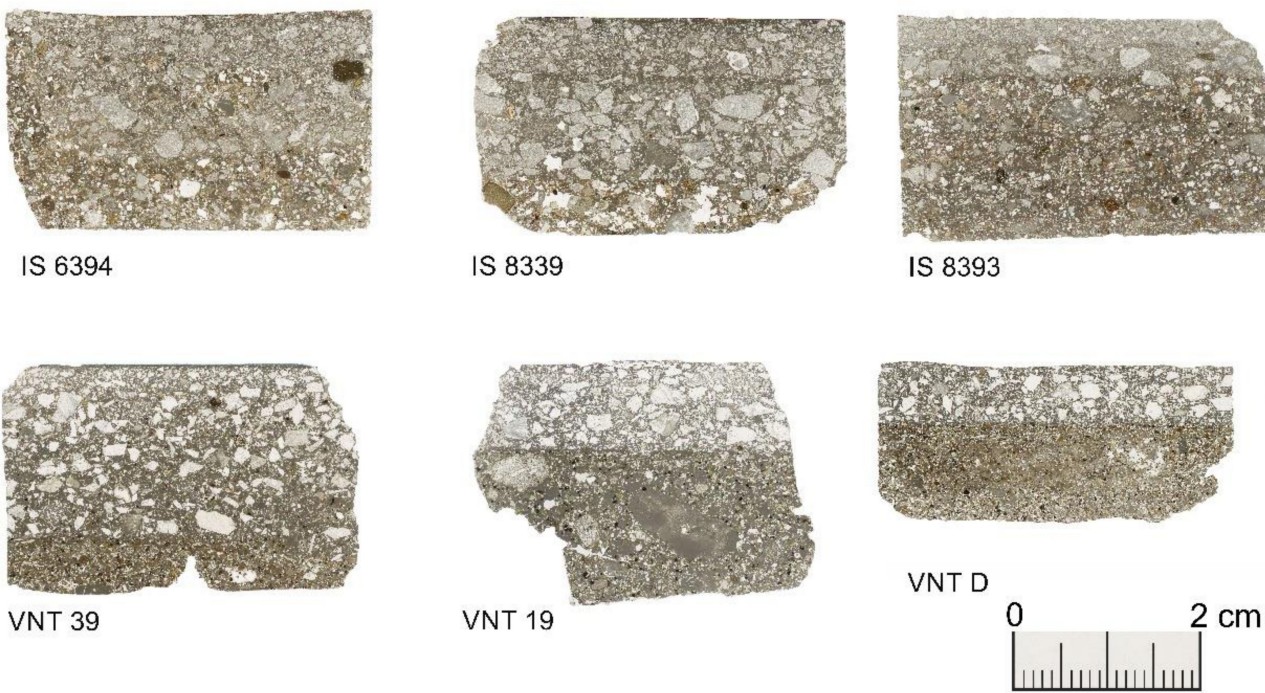

**Figure 2.** Thin sections of the samples IS 6934, IS 8339, IS 8393, VNT 39, VNT 19 and VNT D, under transmitted light.

For a general view of structure, texture and mineral composition, the petrographic thin sections of the plasters were analysed under transmitted light with a Carl Zeiss Jena Jenapol polarising microscope with crossed and parallel nicols (magnification: 3.2×, 10×, 20× and 50×). The petrographic analysis is particularly useful for recognising the type of binder and the nature of the aggregate and to understand the kind of stone used to produce the lime, the ratio between binder and aggregate, the origin and composition of the aggregate, and the supply areas of raw materials used in the realisation of the mortar.

For confirmation of the mineral phases and observation of greatly enlarged details (from 60× to 1000×), optical microscopy was supplemented with an examination under a scanning electron microscope (SEM), model ZEISS EVO 40 XVP, equipped with an energy-dispersive X-ray detector (EDX) in variable pressure modality (camera pressure ~25 Pa), creating backscattered electron images, which were very useful for observing differences in composition. As a matter of fact, the observation in variable pressure mode on noncoated material allows us to analyse thin sections without resorting to coating, which would affect the subsequent use under the optical microscope. The result is a semiquantitative elemental analysis that is enough for the purpose of recognising the mineralogical phases. SEM observation in backscattered mode produces images that emphasise the compositional

differences; this allows us to recognise with greater precision the mineralogic phases, and in particular, those that occur in crystals with dimensions that are difficult to observe under the optical microscope.

M.C., P.F.

## 3. Results

The preparatory layers

In the samples, up to five layers can be recognised (excluding the surface paint layer) with some significant differences between the plaster of the vertical walls and that of the ceilings. In fact, in the first case (IS 6934 and 8339, VNT 39 and 19) there are two or three layers, the most internal of which is grey in colour, between 5 and 15 mm thick, and made of lime and sand, and one or two surface layers that are white in colour, composed of lime and carbonatic fragments, and 10 mm (internally) and 3–6 mm (surface) thick. In the ceiling plasters, however, two or three thinner grey layers made of lime and sand can be recognised, which, overall, reach a maximum thickness of 17 mm (IS 8393), and there are one or two superficial white layers, which are also thinner compared with the previous samples, and are 6–7 mm thick altogether.

The sandy layers in the samples from Isera consist of coarse sand with subrounded/subangular grains ($\phi$ average ~1 mm) composed of carbonate rocks (dolomite, limestone, fossiliferous limestone, calcarenites) and crystalline rocks (metamorphites, granites, volcanites sometimes altered) in a fine calcium carbonate binder, locally in patches with a micritic texture and pores of irregular morphology with forms of secondary crystallization. On the other hand, the samples from Ventotene have rounded/subrounded and homogeneous sand ($\phi$ average ~0.5 mm, but VNT 19 also contains grains reaching up to a centimetre in size) characterized by a mixed composition of bioclasts (molluscs, foraminifera, echinoderms, bryozoans) and volcanic products (including monocrystals of their phases, especially pyroxenes) (Figure 3). Locally, the microliths show edges of alteration; the mixture is poor and the particles of mineral aggregates are embedded in a brown binding agent characterized by a nonhomogeneous aspect with a micritic texture, pores, and crystallized microfissures with irregular distribution and form (Figure 4).

The carbonate layers are even and uniform; the filler is composed of fine angular/subangular gravel ($\phi$ from 2 to 5 mm) embedded in a calcium carbonate binder (Figure 5). In the samples from Isera we are dealing almost exclusively with microcrystalline dolomite (Figure 6), whereas in those of Ventotene there is cleavage calcite, and in smaller measure, calcareous rock (micrite). In all the samples, porosity and cracking are practically absent. When two layers are present, the top one is usually finer and thinner, with well sorted grains of a smaller size ($\phi$ average ~1 mm); those of a more elongated shape have an orientation parallel to the surface.

The pigments

The paint layer is clearly distinguishable in three samples. The SEM analysis permitted the determination of the nature and method of applying the pigments. Specifically, the black colour on one of the samples from Isera (IS 6934) results as being composed exclusively of carbon, and as such, is interpretable as carbon black or black vine (Figure 7); the paint layer, between 100 and 200 μm thick, has been applied on a preparatory layer with an irregular surface. A second sample from Isera (IS 8339) has on the surface a layer of crushed mercury sulphide (HgS = cinnabar) (Figure 8); the fragments reach up to 20–25 μm; the paint layer, at its thickest depth of 70–80 μm, is applied on a base reaching a maximum thickness of 40 μm and composed of extremely fine microcrystalline mortar. The third sample from Ventotene (VNT 39) was found to be composed of angular, elongated fragments reaching a maximum size of 70 μm of calcium copper tetrasilicate ($CaCuSi_4O_{10}$ = cuprorivaite) mixed with quartz and amorphous silica, that is, Egyptian blue frit (Figure 9); the paint layer has a regular thickness of 300 μm and is applied directly on a perfectly levelled and smoothed preparatory layer.

M.C., P.F.

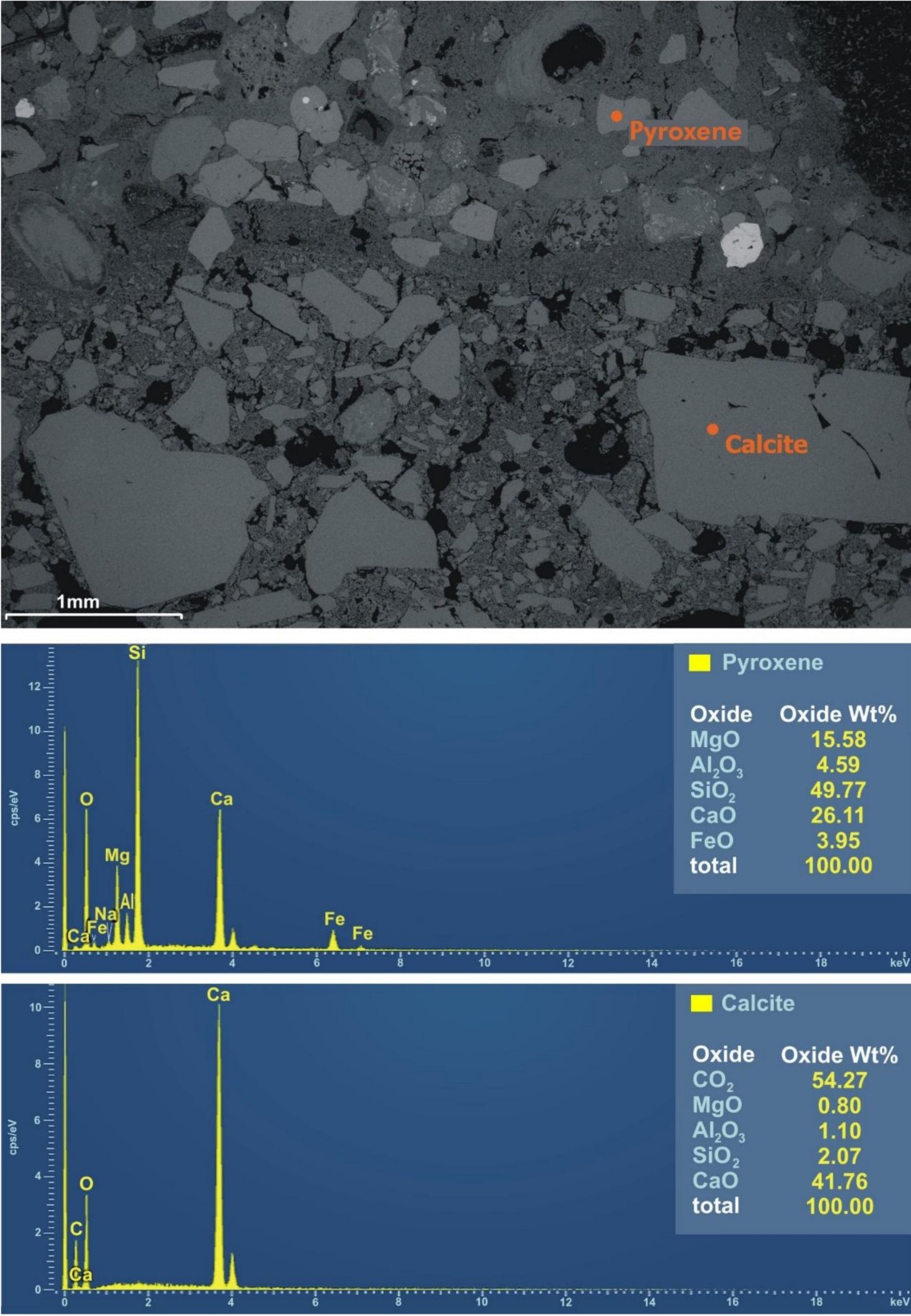

**Figure 3.** Backscattered SEM image of sample VNT 19, highlighting the border between a layer rich in bioclast volcanoclastic elements (**top**) and a layer with cleavage calcite grains (**bottom**). To the right, EDS spectra and tables of elemental quantification of a pyroxene grain and a calcite grain.

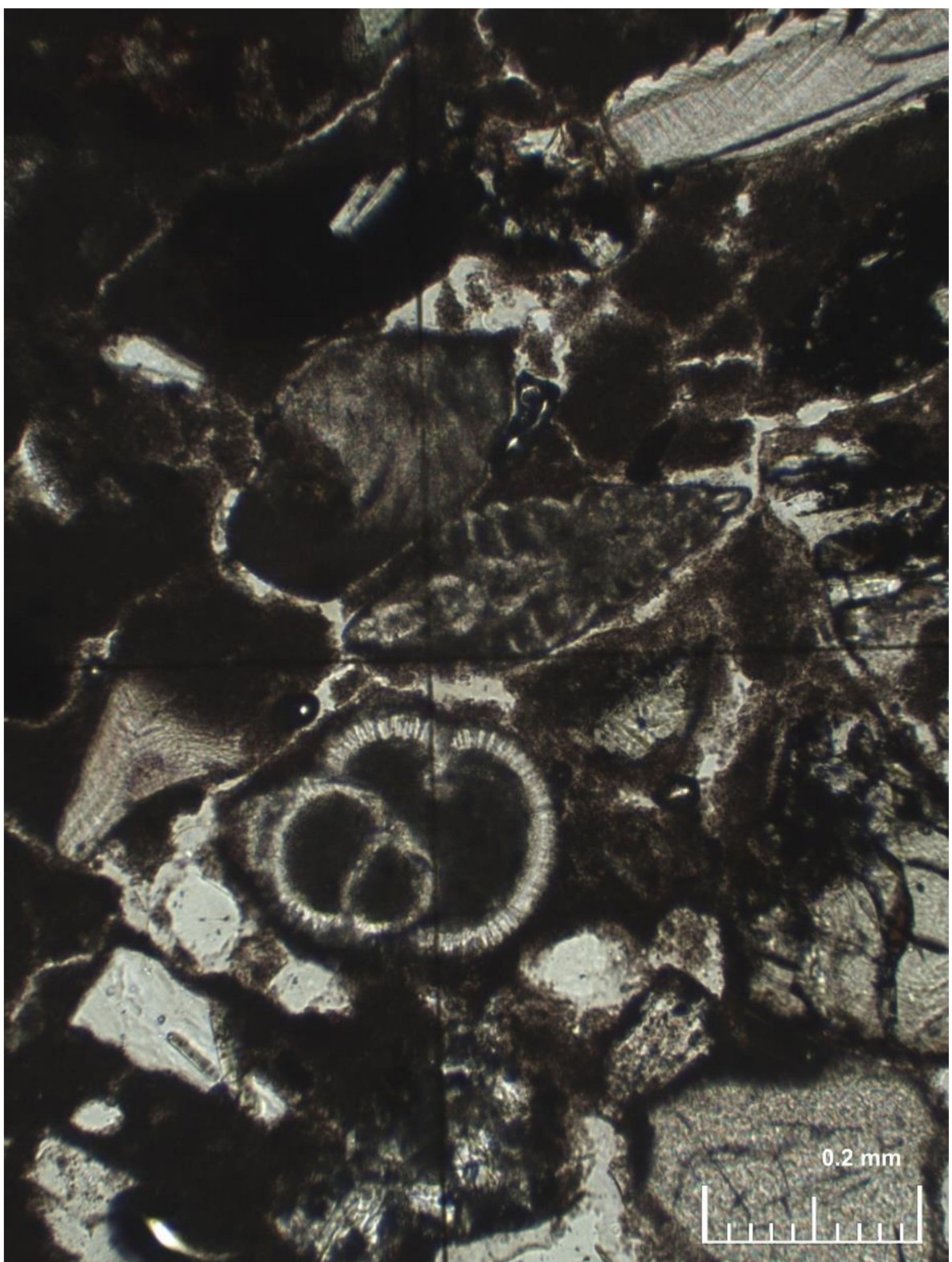

**Figure 4.** Thin-section VNT 39 under polarizing microscope (parallel nicols), magnification 10×. Note the lean paste, the brown colour of the binder and the abundance of bioclasts.

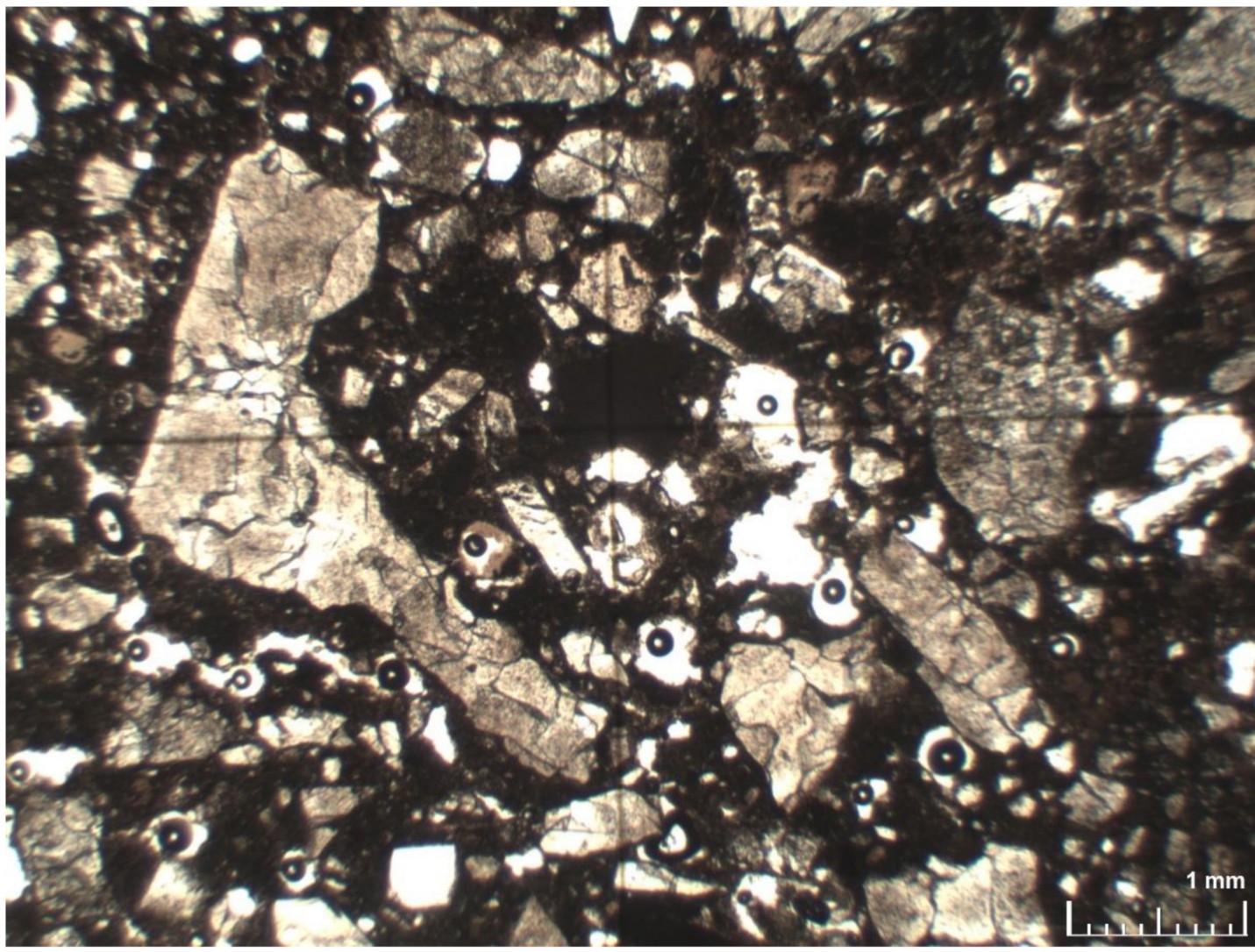

**Figure 5.** Thin-section IS 6934 under polarizing microscope (parallel nicols), magnification 3.2×. Note the angular/subangular shape of grains consisting almost entirely of microcrystalline dolomite in a fine calcium carbonate binder.

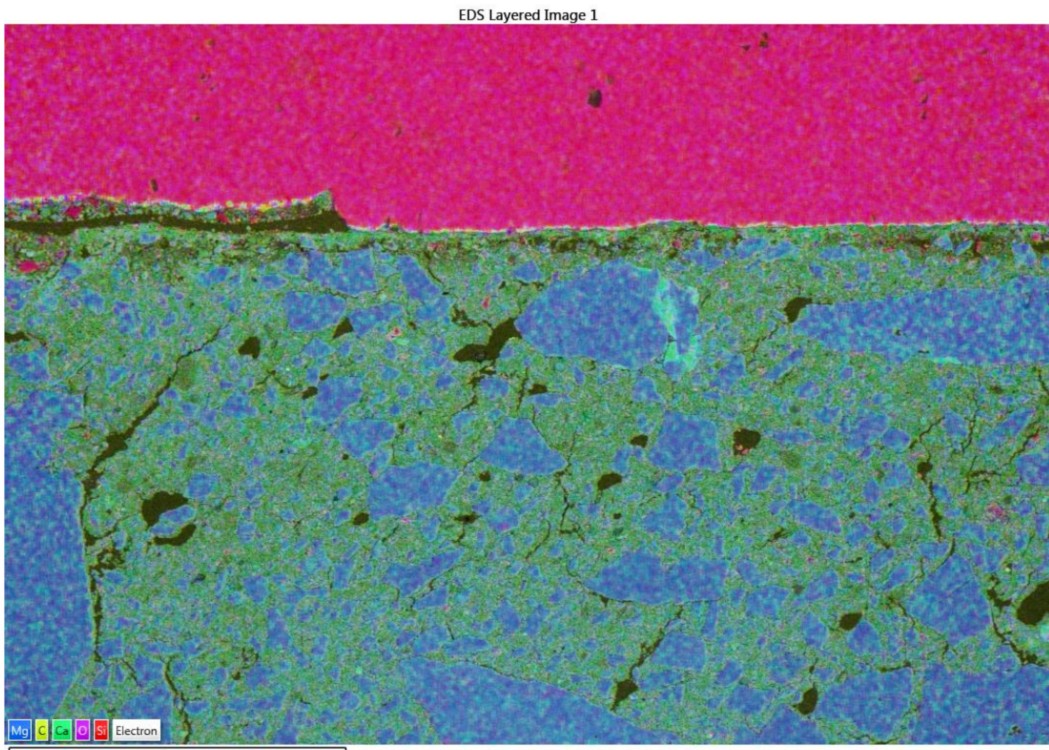

**Figure 6.** SEM-EDS compositional map in false colours of the top preparatory layer of sample IS 6934. Blue = magnesium (contained in dolomite grains); green = calcium (predominant in calcium carbonate plaster).

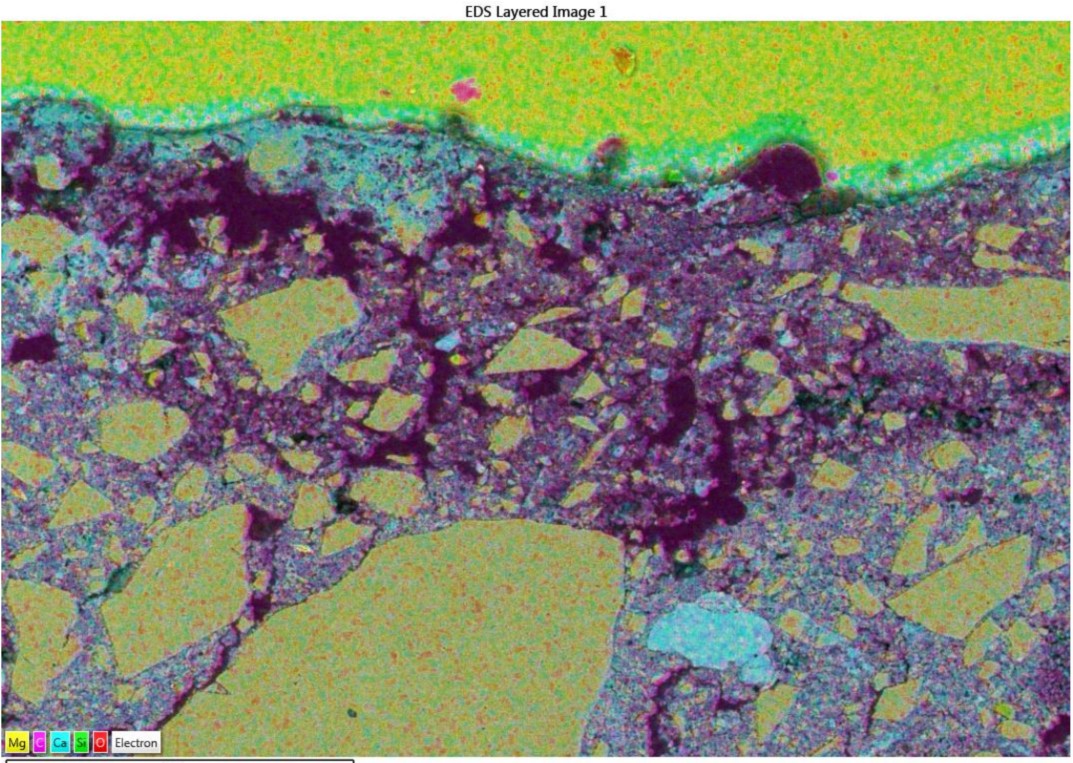

**Figure 7.** EDS compositional map in false colours of the paint layer of sample IS 6934. The purple hues indicate the presence of carbon that impregnates the outer layer (thickness about 200 μm).

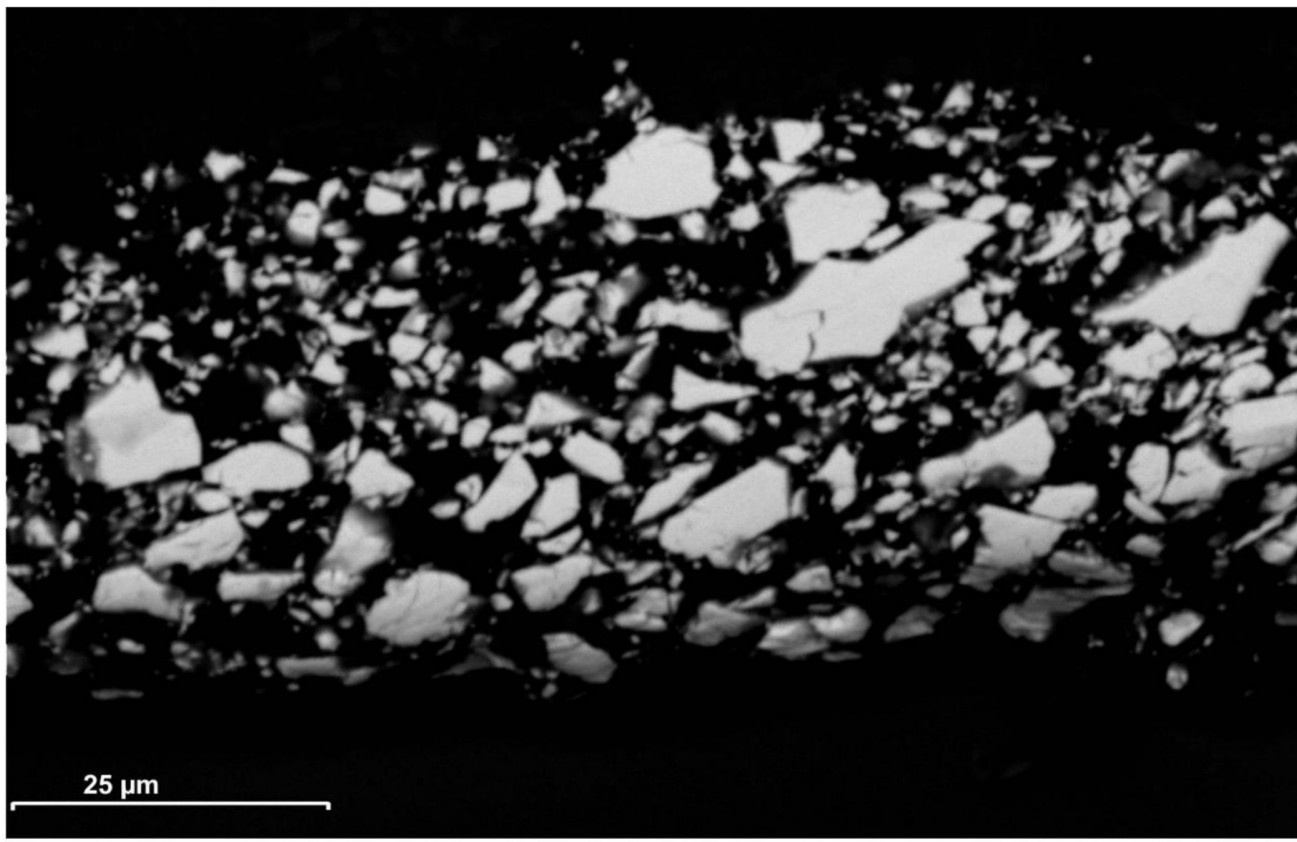

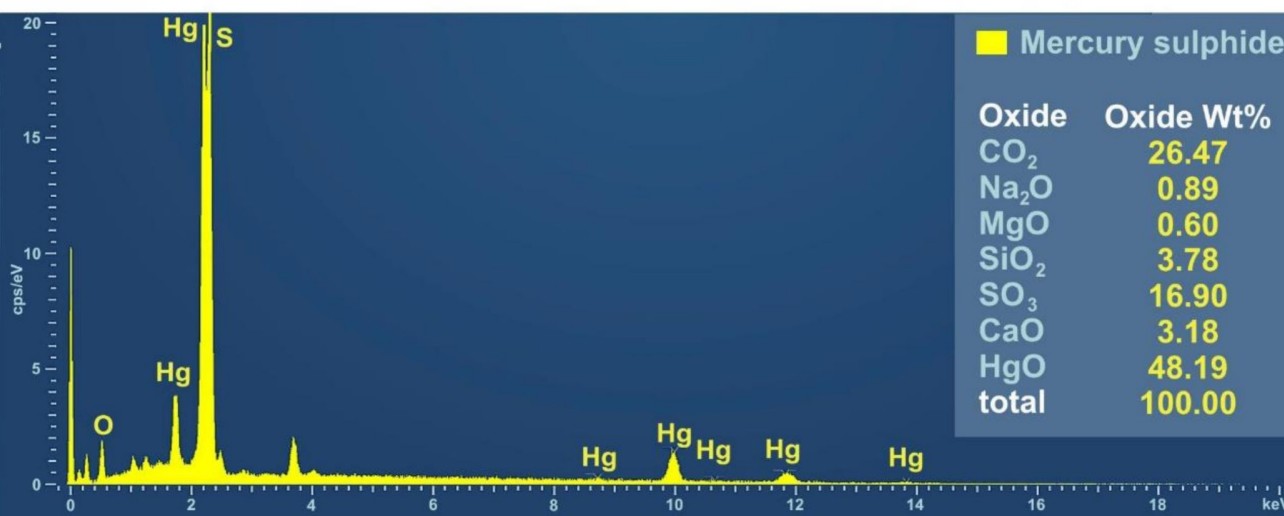

**Figure 8.** Backscattered EDS image of the paint layer of sample IS_8339 with fragments of mercury sulphide (in white). To the right, EDS spectrum and table of elemental quantification of a fragment of mercury sulphide (HgS).

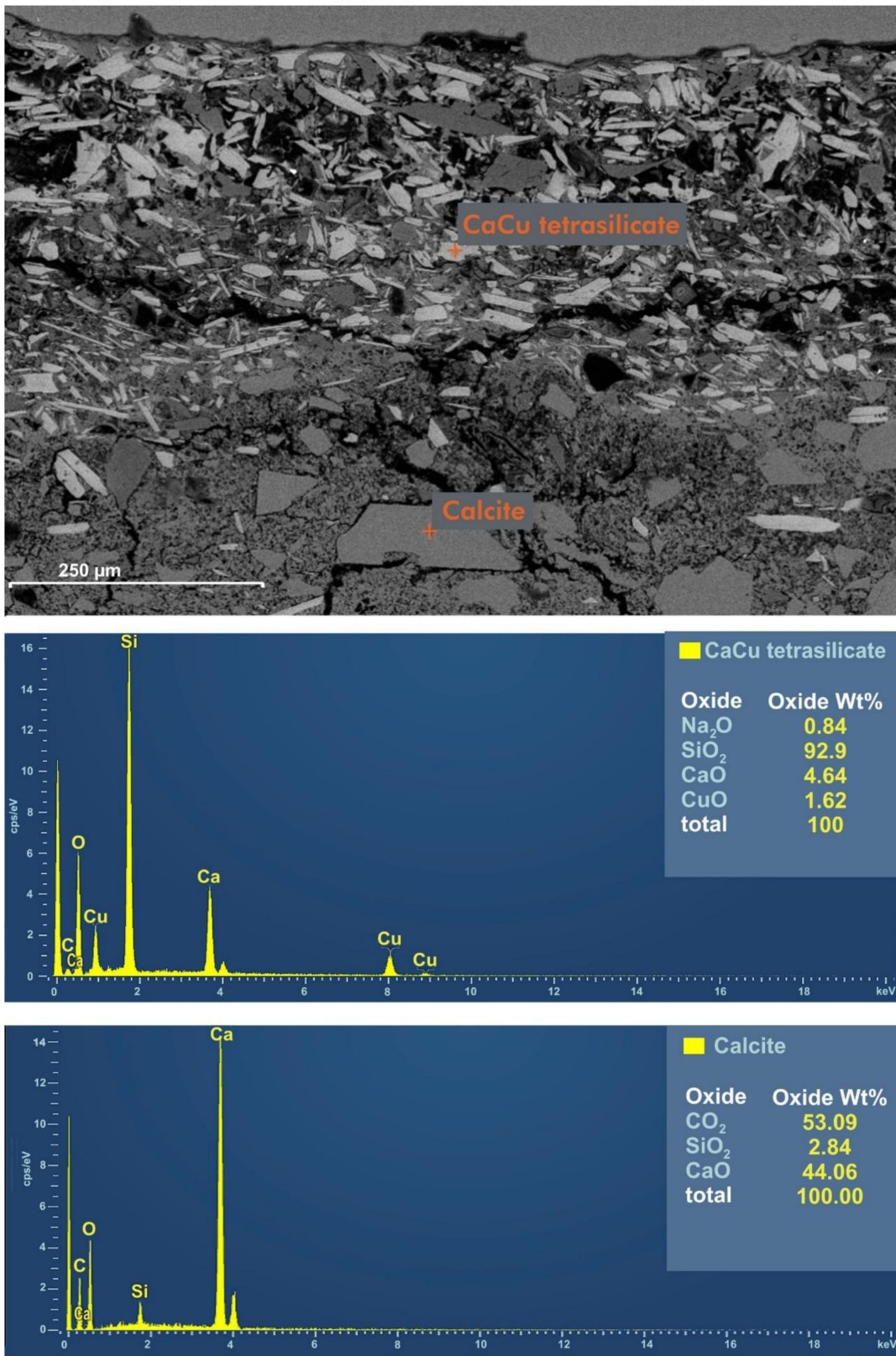

**Figure 9.** Backscattered EDS image of the paint layer of sample VNT 39. Note the presence of angular, elongated fragments of calcium copper tetrasilicate (in white) mixed with minor amounts of quartz and amorphous silica (in grey). To the right, EDS spectra and table of elemental quantification of a calcium copper tetrasilicate fragment ($CaCuSi_4O_{10}$) and of a calcite grain $CaCO_3$).

## 4. Conclusions

The data derived from the comparative archaeometric analysis revealed substantial similarities in the techniques of execution of the wall coverings in the Roman villas of Isera and Ventotene (Table 1). In all cases, one or more preparatory layers of mortar made of lime and mixed sands were covered with one or two white layers of plaster composed of lime and carbonatic material. Both at Isera and Ventotene, the ceiling plasters were found to be composed of thinner layers than those of the vertical walls. The quality of the preparation is very high, especially when compared with examples from the Middle and Late Imperial Age, when the number and the thickness of preparation layers seem to decrease and the microcrystalline material in the top layers tends to disappear [4,10–12].

**Table 1.** Main characteristics of the analysed samples.

| | IS 6934 | IS 8339 | IS_8393 | VNT 39 | VNT 19 | VNT D |
|---|---|---|---|---|---|---|
| Paint layer | Carbon (100–200 μm) on a fine micro-crystalline mortar (1–1.5 mm) | Cinnabar = HgS (300 μm) on a fine microcrystalline mortar (40 μm) | | Calcium copper tetrasilicate, quartz and amorphous silica = Egyptian blue frit (70 μm) | | |
| White layers | Two layers (tot. 15 mm) of lime and fragments of dolomite | Two layers (tot. 17 mm) of lime and fragments of dolomite | Two layers (tot. 7 mm) of lime and fragments of dolomite | Two layers (tot. 18 mm) of lime and fragments of calcite | One layer (10 mm) of lime and fragments of calcite | One layer (7 mm) of lime and fragments of calcite |
| Grey layers | One layer (8 mm) of lime and sand (50% carbonatic clasts; 50% metamorphic and volcanic clasts) | One layer (6 mm) of lime and sand (60% carbonatic clasts; 40% metamorphic and volcanic clasts) | Three layers (17 mm) of lime and fragments of sand (50% carbonatic clasts; 50% metamorphic and volcanic clasts) | One layer (5 mm) of lime and sand (55–60% carbonatic bioclasts; 40–45% volcanic products) | One layer (16 mm) of lime and fragments of sand (30% carbonatic bioclasts; 70% volcanic products) | Two layers (10 mm) of lime (50% carbonatic bioclasts; 50% volcanic products) |

Differences between the two sites can be noted in the materials used and are due to the supply in different geological contexts. In particular, the sandy layers in the samples of Isera contain grains of carbonate and "crystalline" rocks (metamorphites, granites, volcanites) traceable to deposits of glacial/fluvioglacial origin of the Adige glacier, which are widely available in the area. At Ventotene, on the other hand, the rounded fragments made up of marine bioclasts and volcanic products characterised by a composition compatible with the volcanism of the island suggest the use of local marine sand. For the superficial preparatory layers at Isera, use was made of clasts of microcrystalline dolomite, a lithotype outcropping along the Adige valley not far from Isera and in the nearby side valleys, whereas at Ventotene, cleavage calcite was used, which did not come from the island but probably from somewhere on the Italian peninsula. As regards the samples from Ventotene, no compositional difference was found between the preparation of the painted plaster and that of the stucco decoration.

As for the pigments analysed, black, red cinnabar and Egyptian blue, these were some of the most popular colours in the Roman period and, in particular, in the First Imperial Age, as testified by Vitruvius (7, 7–14) and Pliny the Elder (*nat*. 35, 29–50). In particular, the carbon black (*atramentum*), made of carbonized vegetal material, was widely used in the third style paintings to decorate *triclinia* and prestigious rooms [13,14]. The cinnabar (*minium*), composed of mercury sulphide, was one of the most expensive colours in antiquity [15]; at Isera it was used substantially pure, but, unlike the black, seems to have been applied to a very thin layer of microcrystalline plaster, which was probably applied in order to level the surface of the preparatory layer and, thus, avoid wasting precious pigment. Equally precious in the Roman period was the Egyptian blue frit (*aegyptium*), a synthetic pigment produced from a mixture of copper, lime, quartz and silica [16]; in the

sample from Ventotene, this is applied in quite a thick layer, and uniformly distributed on the surface of a perfectly smooth plaster. The generous quantity of pigment, which seems to have been applied without heeding the cost, is justified in this case by the particularly elevated status of a customer of imperial rank.

P.F., M.C., B.M.

**Author Contributions:** Conceptualisation, B.M.; methodology, P.F. and M.C.; investigation, P.F. and M.C.; writing—original draft preparation, P.F., M.C. and B.M.; writing—review and editing, B.M.; supervision, B.M. All authors have read and agreed to the published version of the manuscript.

**Funding:** The project did not receive specific grant from any funding agency in the public, commercial or not-for-profit sectors.

**Institutional Review Board Statement:** This study did not require ethical approval.

**Informed Consent Statement:** This study did not involve humans.

**Data Availability Statement:** The data presented in this article are available on request from the authors.

**Conflicts of Interest:** The authors declare no conflict of interest.

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
