# Peer review of "Archaeometric Characterization of Wall Paintings from Isera and Ventotene Roman Villas"

_heritage, doi:10.3390/heritage5040170_

Round 1
Reviewer 1 Report
The paper deals with an attractive subject, comparing plaster composition found in samples obtained from two Roman villas located in very different locations of the Italian peninsula. In the technology behind these plasters, they find some interesting similarities, while also finding differences mainly related with the use of some locally available materials.
The English in the text is good and can be understood without many complications. I would just give it another read to correct or improve some areas of the text.
I would recommend the paper for publication only after the text (and mainly the introduction) is enlarged to include some basic information that the reader requires to better appreciate this paper.
Introduction:
The introduction is too short and missing important information. Some recommendations that would greatly help enrich this section would be:
- Why is it important to carry out this comparative study of plasters from different parts of Italy. Additionally, is this an ongoing study where plaster from other parts of Italy will be studied, or is it limited to the two villas presented in this paper? If this is part of a wider study of many plasters from different parts of Italy, please quickly introduce the reader to this wider study, including if it is limited to Roman villas, is it for a single era (e.g. First Imperial Age), or for different Roman times. If the study is only of the two villas presented here, please change your text to reflect this; for example, instead of saying from different parts of Italy, it would be better to be more specific and say that the comparison is of two villas located in central and northern Italy.
- Why were these two villas selected to be presented in this paper.
- Please at least briefly introduce the reader to plasters and their study. It is also necessary to include a discussion and references of previous studies on Roman plasters. This would allow comparing your results with those from other areas and/or time periods, and allow for a much richer discussion than that obtained with just two samples.
- Introduce the reader to the methodology that will be followed in the analysis of these wall plasters.
Materials and methods:
- The description of the analyzed samples, which is found at the end of the introduction, should be moved to this section.
Even though the description of where these samples were acquired may be sufficient, providing a little more detail of their origin, and more importantly a justification of why they were selected, may help understand the discussion of the results.
- Page 3, line 44. please correct "... useful for observing compositional differences."
Results
- Page 3, line 49. The phrase in parenthesis "(not preserving the complete sequence of preparatory layers but just the most superficial part of the rendering)" is confusing. Please consider rewriting this sentence to make it clearer.
- Figures. 3, 8, 9. The EDS spectra presented are way too small. I would also argue that it isn't necessary to include these spectra, and it would be more useful to provide a small table with the elemental quantification values instead (or the elemental intensity if not quantified). If desired, these tables do not necessarily need to be included as a separate object, they could just take the same space as the spectra in their respective figures.
Conclusions
- I would rename the section to "Discussion and conclusions", as this seems more of a discussion of the results than a conclusion.
- This section offers a good description of the comparison between the plasters from both villas. But since the focus of the study is precisely on this comparison, I would suggest adding a table or image that can provide a visual aid for the reader to quickly observe the main results and the similarities and differences that are discussed in the text.
- Finally, as mentioned on my comments for the introduction, it is important to include a justification of why it is important to compare these plasters. This justification would also help enrich the conclusion, as there is no real conclusion in this text, just a discussion of the results obtained.
Author Response
Dear Reviewer,
thank You very much for Your useful observations that helped us improve our manuscript. In particular, based on Your recommdendations we have made the following revisions:
- we enriched the introduction adding concise information about the study of wall plasters and about the purpose of our project, that is still ongoing;
- we added some information about the origin of the samples
- we moved the description of the analysed samples to the section "Materials and methods"
- we have deleted some unclear expressions
- we replaced Figures 3, 8 and 9 with new Figures, bigger and clearer, adding also tables with elemental quantification
- we renamed the Section to "Discussion and conclusion" and we put some brief considerations about the importance of comparing plasters
- we added a Figure/table providing the main results of the analyses
We hope that these changes will meet Your favor and we look forward to a positive response.
Best regards
Reviewer 2 Report
In this article Roman wall paintings plasters from two different sites, Isera and Ventotene Roman Villas, by archaeometric characterization. The topic of the research is interesting, but before it could be accepted for publication, extensive changes are needed and the manuscript should be considerably improved. Therefore, I recommend accepting this contribution after major revision. For more information please see the comments below.
Introduction
(1) The introduction part is too short and insufficient and needs to be expanded. It does not refer to the technologies of wall plastering and wall paintings, nor to parallel archeometric studies that were published in the scientific literature during the last few years. Please explain what kind of methods were used by others in order to characterize plaster and pigments of samples retrieved from ancient decorated walls.
For more information please see:
Arjonilla, P., Domínguez-Vidal, A., Rubio Domene, R., Correa Gómez, E., de la Torre-López, M.J. and Ayora-Cañada, M.J., 2022. Characterization of wall paintings of the Harem Court in the Alhambra Monumental Ensemble: Advantages and limitations of in situ analysis. Molecules, 27(5), 1490.
Ashkenazi, D., Shnabel, R., Achim, L., and Tal, O. 2021. Chemical analysis of plaster and pigments retrieved from a decorated house wall at seleucid Tell Iẓá¹abba (Nysa-Scythopolis, Beth She’an, Israel). Mediterranean Archaeology & Archaeometry, 21(3), 89–122.
Bartolozzi, G., Bracci, S., Sacchi, B., Realini, M., and Mazzei, B., 2021. Mural paintings of the cubicle “dei fornai” in Domitilla catacombs in Rome: a study via non-invasive techniques. Archaeological and Anthropological Sciences, 13(11), 1–11.
Gliozzo, E., Pizzo, A. and La Russa, M.F., 2021. Mortars, plasters and pigments—research questions and sampling criteria. Archaeological and Anthropological Sciences, 13(11), 1–30.
Guglielmi, V., Comite, V., Andreoli, M., Demartin, F., Lombardi, C.A. and Fermo, P., 2020. Pigments on Roman wall painting and stucco fragments from the Monte d’Oro Area (Rome): a multi-technique approach. Applied Sciences, 10(20), 7121.
Ion, R.M., Barbu, M.G., Gonciar, A., Vasilievici, G., Gheboianu, A.I., Slamnoiu-Teodorescu, S., David, M.E., Iancu, L. and Grigorescu, R.M., 2022. A Multi-Analytical Investigation of Roman Frescoes from Rapoltu Mare (Romania). Coatings, 12(4), 530.
Radpour, R., Fischer, C. and Kakoulli, I., 2019, June. New insight into Hellenistic and Roman Cypriot wall paintings: an exploration of artists’ materials, production technology, and technical style. Arts, 8(2), 74.
Salvadori, M. and Sbrolli, C., 2021. Wall paintings through the ages: the roman period—Republic and early Empire. Archaeological and Anthropological Sciences, 13(11), 1-30.
(2) Figure 1: A sign indicating the North direction as well as a scale bar should be added to the map.
(3) The references must be cited within the text according to the heritage, MDPI journal guidelines.
(4) At the end of the introduction, the research methodology and the purpose of the research should be explained.
Materials and Methods
(5) Do not use abbreviations like (MC, PF) without specifying the full concept first.
(6) Information should be added about the preparation of the petrographic samples.
(7) More information should be provided concerning the characterization methods that were used, including the data that can be obtained from each method, the advantages and disadvantages of each method, and the degree of error of the used instrumentation. It could be useful to examine the samples with additional tools such as XRF, Raman spectroscopy, and XRD analyses. For further information, please see the list of recommended references above.
Results
(8) Some of the information in the results part should be moved to the Discussion part.
Discussion
(9) A Discussion part should be added.
References
(10) The references must be numbered and arranged according to the journal requirements.
Author Response
Dear Reviewer,
thank You very much for Your useful comments. Following Your recommendations as much as possible also in accordance with the observations of Your Colleague, we have made the following changes:
- we enlarged and enriched the Introduction adding concise information about the study of wall plasters and we tried to contextualize our project and the purpose of our research also adding some references
- we have added information about the preparation of petrographic samples and about the characterization methods that have been used. We are aware that other methods could be used to collect more data, but at the moment we do not have these possibilities in our museums.
- we renamed the last section to "Discussion and conclusions" (see Reviewer 1), adding some brief considerations regarding the importance of comparing plasters
- we have deleted the abbreviations MC, PF and BM inserting the full names of the authors
- we added new references and we numbered them as required by the Journal
As for Figure 1, by convention the North direction is up and the scale bar seems to be superfluous in the image of an entire state. The photos inserted are intended to give a suggestion of the sites; if inadequate they can be deleted.
We hope that the revision meets Your favor and we look forward to a positive response.
Round 2
Reviewer 1 Report
The article has improved considerably, as the reader can now understand the context much better without having to resort to additional bibliography, which was also lacking in the previous version.
I can now recommend the article for publication. I just have a couple of smaller recommendations that would help improve the text.
As required in the previous review, the authors added a justification of the techniques used, but this was included in the "2. Materials and Methods" section, and divided as individual descriptions of the techniques used, without describing the methodology followed. It is more appropriate to include this description in the introduction (as recommended in my previous revision), introducing the methodology followed and a justification of why this methodology (or the techniques used for the study) was selected for the present study.
In page 13, Figure 10 is a table, and should read Table 1.
Additionally, the table does greatly help visualize the differences between the samples, but it is full of text that makes it a bit hard to read. I would recommend finding a way to simplify the presentation of the results in order to remove much of the text in the table, and make it easier for the reader to identify the differences and similarities found in the samples.
Finally, while not incorrect, the spectra presented in figures 3, 8 and 9 is now a bit too large. I would recommend making the spectra a little bit smaller while preserving the size of the tables with the elemental content.